# Early Mobilization of Critically Ill Patients: A Survey of Knowledge, Practices and Perceptions of Greek Physiotherapists

**DOI:** 10.3390/healthcare13111248

**Published:** 2025-05-26

**Authors:** Elpida Papadimitriou, Stavros Petras, Georgios Mitsiou, Ioannis Vasileiadis, Eirini Grammatopoulou, Irini Patsaki

**Affiliations:** 1Department of Physiotherapy, University of West Attica, 12243 At1hens, Greece; 2Medical School, National and Kapodistrian University of Athens, 12243 Athens, Greece

**Keywords:** early mobilization, intensive care units, barriers, perception, physical therapists

## Abstract

Background/Objective: Early mobilization (EM) of critically ill patients is a feasible and safe intervention that limits the implications of bed rest and improves lung function. However, its limited implementation suggests a gap between the research evidence and clinical practice. It is widely accepted that early mobilization faces a variety of barriers. This study aimed to investigate the perceptions of Greek physiotherapists on EM barriers and record their knowledge and practices. Methods: We conducted an electronic survey using the online platform “Microsoft Forms”, among critical care physiotherapists in 66 hospitals that had an Intensive Care Unit (ICU) department in Greece in 2024. We administered a questionnaire, developed based on valid and reliable international questionnaires, with the following domains: education and knowledge on early mobilization, practices, perception regarding EM, and perceived barriers to early mobilization. Results: A total of 126 Greek physical therapists participated. The majority of them worked in urban area hospitals and in a rotation schedule around all departments. Most physical therapists stated that early mobilization is a priority for the patient’s rehabilitation and an important factor in preventing the complications of bed rest. Yet, they do not use specific protocols. Most had knowledge of what EM involved and the international guidelines. The most common barriers reported were the hemodynamic instability and the incoherence with the ventilator. Dedicated physiotherapists singled out certain barriers like the presence of delirium and the lack of communication among ICU staff. Additionally, physiotherapists with more years of experience did not acknowledge tubes, connections, femoral lines and Body Mass Index (BMI) as barriers. Conclusions: Most Greek physiotherapists believe that early mobilization is crucial for the rehabilitation of critically ill patients. A significant percentage know the guidelines, yet they do not follow a specific protocol. Various barriers prevent its implementation, which depends on the patients, healthcare providers, and the overall process. Yet, It is recognized that practices and perceived barriers are influenced by experience and work schedule. Establishing clinical protocols is essential to facilitate the implementation of early mobilization and support patient rehabilitation. Future efforts should focus on designing strategies and EM protocols for physiotherapy in Greek ICUs. Also, we need to monitor changes in perceived barriers across other countries as focus on the matter via published studies and clinical seminars could lead to significant changes.

## 1. Introduction

Many patients in the Intensive Care Unit (ICU) are often confined to bed rest [1,2]. This immobilization has detrimental effects on their physical and mental functioning, including reduced muscle strength, increased mechanical ventilation duration, and prolonged length of stay in ICU and hospital [1]. Also, the lack of mobilization has been shown to result in up to 40% muscle mass loss in the first week and 1% bone density loss per week [3]. The most reliable intervention for preventing these complications is early mobilization (EM) [2,4]. This is defined “as mobilization within 72 h of ICU admission, which is feasible and well-tolerated by most patients once they are stable” [5]. It involves a progression of a series of activities from passive and active range of motion to full ambulation throughout ICU stay [2]. Many studies have shown that this intervention is safe and feasible in critically ill patients [6]. A variety of benefits of EM have been reported such as improved physical and mental function, reduced ICU length of stay, and quality of life [6].

Despite its potential advantages, EM is not widely implemented [7,8]. Numerous international studies have reported a low prevalence of out-of-bed mobilization, especially among intubated patients on mechanical ventilation [7,8]. For example, a 1-day German point-prevalence study of which included 783 mechanically ventilated patients across 116 ICUs found that only 24% of these patients were mobilized out of bed as part of routine care. Among patients with an endotracheal tube, this number dropped to just 8% [8]. Similarly, a study in southern Brazil revealed that only 10% of all mechanically ventilated patients and a mere 2% of patients with an endotracheal tube were mobilized out of bed [9].

Several studies have examined why EM is not commonly implemented in ICU clinical practice [10,11,12,13]. The gap between research evidence and clinical practice highlights the presence of various barriers [10,11,12]. Previous publications have categorized these barriers to EM into three domains: (1) patient barriers, such as patient symptoms and conditions, endotracheal tubes, monitors and catheters; (2) provider barriers, such as limited human and technical resources and insufficient training; and (3) institutional barriers related to the ICU culture, lack of proper guidelines, lack of coordination, and lack of rules for the distribution of tasks and responsibilities [11,12].

Many studies have investigated these barriers at the local and international levels to identify them and design strategies for their resolution [14,15]. In Greece, ICU physiotherapists practice both EM and respiratory techniques [16,17]. A recent study found that while 50% of participating ICU physiotherapists were aware of early mobilization guidelines, only 19% reported implementing them [16]. To date, there has not been a study examining the barriers to implementing early mobilization which act as inhibitors in promoting the recovery of critically ill patients in the ICU [18]. The primary aim of the present study was to identify physiotherapists’ knowledge and practices regarding EM. It was also to identify their perceptions of barriers to EM in Greek ICUs. The secondary aim was to examine whether these results have been influenced by participants’ years of work experience or dedication in the ICU.

## 2. Methods

### 2.1. Study Design: Survey Development

This was an electronic survey study that was conducted from May 2024 to November 2024. The questionnaire used was developed by a panel of experts who systematically reviewed similar published studies [10,11,12,13,14,15,18,19,20,21,22]. The panel comprised one registered nurse and four physiotherapists, all with a PhD and extensive experience in the ICU setting. The questionnaire that was developed was divided into the following domains, which focused on: participants’ knowledge of early mobilization (6 items), practices (7 items), perception on EM (11 items), and perceived barriers (29 items) to its implementation. Before these domains, we also included four questions regarding sociodemographic variables such as: sex, age, residential area (urban, rural, etc.), and educational level. Considering the barriers identified by the initial systematic review of literature, we divided them into the following categories: patient-related, institutional, provider-related, and cultural barriers. The survey included only closed-ended questions. The format of the responses varied, but mainly included: “Strongly agree-Agree”, “Neutral”, “Disagree-Strongly disagree”, and “Yes”, “Don’t Know”, and “No”.

The questionnaire was developed as an electronic survey using the online platform “Microsoft Forms”. To evaluate comprehensiveness and identify any unclear items that would need modification, we emailed the link of the e-questionnaire to thirty physiotherapists with long service in the ICU in Athens and Thessaloniki. This pilot study pointed out a few syntax issues that were corrected after discussing them among the panelists and consensus was reached.

### 2.2. Survey Administration: Data Selection

In order to identify physiotherapists working in the ICUs, we created a list of all Greek hospitals from the main webpage of the Health Ministry, and contacted them to identify which of them had an ICU department and to gather all contact information for the physiotherapy departments. We contacted the head physiotherapists of each hospital, and sent the questionnaire link to the departments’ email addresses. However, the exact number of ICU physiotherapists could not be determined due to the absence of an official registry, as most physiotherapists are not exclusively employed in Greek ICUs. Additionally, the questionnaire was posted via social media and in special interest groups related to ICU physiotherapy and ICUs. All Greek physiotherapists that had experience and have worked in the ICU were requested to complete the survey.

Ethical approval was obtained from the ethics committee of the University of West Attica (41564/26 April 2023). Participants completed the survey voluntarily. Also, their anonymity and confidentiality were assured via the online platform. Along with the questionnaire, we included a covering letter explaining the purpose of the study, identifying the corresponding researcher, assuring privacy, and describing the process of handling their responses and any personal data.

### 2.3. Statistics

The SPSS Statistics (Version 21, IBM, Armonk, NY, USA) for Windows software was used for descriptive frequency analysis. Descriptive statistics were used to summarize the responses for all items, along with bar graphs to present our findings.

## 3. Results

### 3.1. Participants

A total of 126 physiotherapists working in Greek ICUs participated in the survey. Among participants, (a) 65% were women, (b) 42% had obtained a master’s degree, and (c) 44% (the largest percentage) were 40–49 years old. Furthermore, 58% of the participants had more than seven years of experience working in the ICU. Most physiotherapists worked on a rotational basis across different hospital departments, with only 20% dedicated solely to the ICU. The largest percentage of participants (43%) noted that this rotation was performed every 3 months. Additionally, 87% of respondents were employed at public hospitals located in urban areas, 75% noted that they did not have a daily afternoon shift, and 62% had a weekend shift. The majority of our ICUs are multidisciplinary, with less than 15 beds. The physiotherapists to patient’s ratio in most cases was 1 to 10–15 patients, depending on the total number of ICU physiotherapists and the presence of more than one ICU department.

### 3.2. Knowledge (Appendix A)

According to the data, 38% of participants received training from a senior colleague, while a similar percentage (30%) attended training seminars, workshops, and relevant conferences (Figure 1). Most of the physiotherapists (94%) reported being aware of the benefits of EM. Approximately two-thirds of the respondents indicated they were familiar with the safety criteria for initiating (70%) and discontinuing (75%) EM. Although 81% knew what EM involved, 39% said they lacked sufficient understanding of the European Society of Intensive Care Medicine guidelines for preventing post-ICU syndrome.

### 3.3. Practices (Appendix A)

Among the physiotherapists surveyed, 71% reported that they do not follow any specific protocol for implementing EM, yet 55% noted following clinical guidelines on the matter. Additionally, 25% indicated they do not utilize the ABCDEF bundle, and this could potentially be higher, considering that 44% of participants were unable to provide an answer. Furthermore, 67% of the participants do not take part in the morning physicians’ briefing, while 86% wait for the physician’s referral before implementing EM.

### 3.4. Perceptions (Appendix A)

Most physiotherapists (85%) believe that EM is a top priority for patients’ rehabilitation and a crucial factor (99%) in preventing complications from immobility. Additionally, a similar number of participants (92%) agree on the importance of having a protocol for the safe implementation of EM. Furthermore, a significant majority (96%) support the idea of using a bundle of practices like the “ABCDEF” framework. The participants (73%) feel that physicians are in favor of EM of patients, but are not sure whether the nursing staff share the same opinion.

### 3.5. Reported Barriers Regarding the Implementation of EM

Although physiotherapists believe that intensivists are in favor of EM—with 73% supporting this view—several barriers to EM have been identified and reported. The most perceived barriers related to patients include hemodynamic instability (94%) and difficulty with ventilator synchronization (61%). In contrast, most responders disagreed that the endotracheal tube (81%), muscle weakness (79%), and poor nutritional status (56%) posed barriers to EM. Additionally, a significant proportion of physiotherapists emphasized the importance of training (94%), and having an EM protocol (92%) or clinical guidelines (96%) are important. Consequently, the lack of these resources is seen as an institutional and provider-related obstacle to facilitating EM. Regarding the equipment used in ICUs, opinions were divided: 36% of physiotherapists agreed on its adequacy while 40% disagreed. Moreover, most respondents reported that the requirement for a physicians’ order before mobilization (86%) is the most frequent process-related barrier. Lastly, the potential of musculoskeletal self-injury was also recognized as a barrier by 76% of physiotherapists. All answers regarding perceived barriers on EM are presented in Appendix A.

### 3.6. Work Schedule Variances

We observed variances in Greek physiotherapists’ knowledge, practices, and perceived barriers related to their ICU work schedules. As mentioned earlier, only 20% of the participants were dedicated solely to the ICU, while 80% worked on a rotation basis across different hospital departments. A significant percentage of participants (43%) indicated that this rotation occurred every 3 months. Among those dedicated to the ICU, the majority also covered shifts on weekends and public holidays (80%) compared to only 57% of those who did not work exclusively in the ICU.

Regarding their knowledge, most participants (80%) included in the survey stated that they were aware of what EM involved. Interestingly, when asked if they knew the criteria for initiating and terminating the procedures, the dedicated physiotherapists up to 84% gave a positive answer in relation to others whose positive answers reached 73%.

Concerning EM practices, 60% of the physiotherapists dedicated to the ICU reported participation in morning briefings with doctors, compared to only 24.8% of the physiotherapists on a rotational schedule. Additionally, up to 48% of dedicated ICU physiotherapists reported keeping a daily record of the patient’s mobilization plan, whereas only 36% followed a protocol for EM. In contrast, physiotherapists on a rotational basis recorded this information up to 38% of the time, with only 19.8% utilizing an EM protocol. Lastly, 84% of physiotherapists who are not dedicated to the ICU reported initiating EM only after receiving written commands from the intensivist, compared to 60% of the dedicated physiotherapists.

As to perceived barriers to EM (Figure 2), 34% of dedicated physiotherapists noted delirium as a barrier, compared to 73% of their counterparts. Similarly, regarding Body Mass Index, 20% of dedicated physiotherapists identified it as a barrier, compared to 27% of others. Additionally, none of the dedicated physiotherapists recognized the presence of an endotracheal tube as a barrier, while 7% of the other physiotherapists did. The presence of a femoral line was seen as a barrier by 9% of dedicated physiotherapists compared to 4% of others; for sedation, these figures were 40% versus 50%. Moreover, when asked to identify hemodialysis sessions as barriers, dedicated physiotherapists provided more positive responses, with 44% indicating it was a barrier compared to 34% of others. In terms of patient refusal being viewed as a barrier, 47.5% of physiotherapists working on a rotational basis agreed with this statement, while only 16% of dedicated ICU physiotherapists shared the same opinion. Notably, an increased percentage of dedicated physiotherapists (40%) did not provide an answer to this question, compared to 20% of their counterparts. 

In relation to the equipment necessary for implementing ΕΜ in the ICU, 16% of dedicated physiotherapists indicated that the available equipment is sufficient, whilst 40% of their non-dedicated colleagues shared this view. Furthermore, when examining barriers such as shift duration and the number of physiotherapists available, we found that a greater percentage of dedicated physiotherapists were unable to provide a clear response (32%, compared to 15% for shift duration, and 20% compared to 10% for adequate physiotherapists).

### 3.7. Years of Experience in ICU Variances

As mentioned earlier, 58% of the participants had over 7 years of work experience in the ICU, while 24% had less than 3 years of experience, and the remaining 18% had between 3 and 7 years. Regarding their knowledge of the ESICM guidelines for preventing post-ICU syndrome, 49% of the physiotherapists with more years of experience responded “Yes”, compared to 35% of those with less experience. Additionally, physiotherapists with more years of experience were more likely to provide positive responses regarding their knowledge of initiating, terminating, and identifying contraindications for early mobilization practices (Figure 3).

When considering patient-related barriers, physiotherapists with more than seven years of experience reported a lower percentage of acknowledgment for certain issues, such as delirium (38%), pain (37%), severity of illness (30%), fatigue (27%), Body Mass Index (18%), lack of motivation (14%) invasive lines (3%), endotracheal tubes (4%), and femoral lines (3%) compared to those with less than seven years of experience (Figure 4). Physiotherapists with less than three years of experience seem to encounter more challenges with patients who are in pain (53%) and those who exhibit a lack of motivation (30%).

In the categories of barriers related to institutional factors and healthcare providers (Figure 5) those with more than seven years of experience showed a higher percentage of agreement in recognizing safety issues (80%) and delays in decision-making (55%) regarding the initiation of EM. Furthermore, a greater number of these experienced physiotherapists believed that the lack of guidelines (65%) and the ABCDEF bundle (41%) represent a significant barrier to EM. In contrast, physiotherapists with less than three years of experience identified certain institutional barriers as significant obstacles. Specifically, they cited issues such as the necessity of a physician’s order to mobilize patients (97%), a limited number of physiotherapists (67%), time constraints (63%), and insufficient equipment (43%).

## 4. Discussion

This is the first survey in Greece aimed at assessing physiotherapists’ knowledge, practices, and perceptions regarding the barriers to early mobilization. This study holds particular significance, as it was conducted after the COVID-19 pandemic, during which many physiotherapists were employed or reassigned to work in the ICU. Most Greek physiotherapists know what EM involves and the safety issues regarding its implementation. Increased knowledge and sufficient experience appear to be responsible for the fact that most Greek physiotherapists identified hemodynamic instability, ventilator incoherence, and the presence of delirium as the most significant barriers to the early mobilization of ICU patients. They also indicated that the endotracheal tube, femoral line, and muscle weakness were no longer considered barriers to implementing EM. Years of experience and work schedule play a significant role in variances noted in practices and perceived barriers. It seems that dedication to working in the ICU could possibly be a solution for overcoming the barriers noted in previous studies.

Our study aligns with other research, particularly regarding the perception of hemodynamic instability as a significant barrier to EM [13,14,15,23,24,25,26]. Additionally, barriers such as sedation and disconnection from devices, particularly the endotracheal tube, have also been reported in other studies [10,12,13,14,15,20,21,22,24,25,26,27]. Yet, from our findings, Greek physiotherapists seem confident in mobilizing patients alongside their monitoring equipment and lines, likely due to their experience and training. It is important to note that most of the studies referenced were conducted before 2020, a time when early mobilization was still in its early stages of implementation. Consequently, Greek physiotherapists are now more informed about how to overcome these barriers, as there are published studies outlining strategies to address them. Moreover, it is worth mentioning that the Panhellenic Physiotherapists’ Association Section of Cardiovascular and Respiratory Physiotherapy—Rehabilitation often organizes training sessions on ICU physiotherapy and EM. The importance of training in overcoming barriers is well presented in previous related studies [21,25,28,29,30,31]. Thus, in a survey conducted by Grammatopoulou et al. (2017), 59.2% of Greek physiotherapists indicated that they had received training to mobilize critically ill patients out of bed [17]. Furthermore, the necessity for protocols and guidelines is recognized by foreign participants as well, with many emphasizing their absence [13,14,15,20,21,22,23,24,27,28,29,30]. This suggests that both Greek and foreign physiotherapists understand the importance of organization and structured procedures. The existence of protocols facilitates effective communication between physiotherapists and ICU staff, contributing to improved time management and coordination of various procedures within the ICU.

Many studies have identified inadequate equipment and a shortage of physiotherapists as significant barriers to implementing EM [13,14,15,20,21,23,24,25,26,27,28,29,30]. Additionally, musculoskeletal injuries have been linked to a lack of staff and equipment [10,13,14,15,21,27,31]. The varied responses from Greek physiotherapists can be attributed to individual differences in work schedules and years of experience. Most Greek physiotherapists do not use equipment outside of the rehabilitation department, so they may not recognize its necessity as described in relevant literature [16]. Limited time is a common concern among all participants, which may stem from ambiguous roles and responsibilities that contribute to increased workloads and insufficient time for patient care. It is also worth noting that many studies involved healthcare workers from various specialties. Moreover, foreign physiotherapists highlighted that the lack of instructions and referrals from physicians significantly hinders the process. Conversely, Greek physiotherapists often take the initiative to implement EM independently, probably as a means to save and organize better their time.

When examining differences in work schedules, it is important to consider that most Greek and foreign physiotherapists reported not working exclusively in the ICU. However, those who do work solely in the ICU tend to achieve better patient management and are more successful in overcoming recognized barriers related to EM. In particular, issues of communication and time management are addressed more effectively when healthcare providers spend their entire shift in a specific department. Having dedicated physiotherapists in ICU has been presented as a facilitator to increase EM of ICU patients [21]. Nevertheless, this approach does not resolve the significant staffing shortages that exist in all healthcare systems.

Differences were observed when we examined the role of extensive experience in the ICU. The ICU has unique demands, and physiotherapists in this setting need a range of skills that are often not covered in their basic training. The European Respiratory Society’s respiratory physiotherapy task force has developed a new harmonized postgraduate curriculum for training in respiratory physiotherapy. This curriculum builds on the core syllabus for postgraduate training and includes many of the skills necessary for working in the ICU [32]. However, we have noted significant differences that may also be attributed to the fact that a portion of physiotherapists with less than seven years of experience were employed or transferred to work in the ICU during the COVID-19 pandemic. Surprising differences that we noticed in knowledge related to years of experience were not shared by the study of Jolley et al. [21].

The majority of Greek participants expressed that collaboration with ICU staff during EM is feasible. However, they were unclear about whether there is effective communication, coordination, and organization regarding the therapeutic interventions that should be implemented for each patient among the ICU team. This suggests that a common communication framework among staff members is lacking, likely due to the absence of established protocols. In contrast, the foreign participants identified the inability to collaborate among staff as a significant barrier to implementing EM effectively [13,14,15,20,22,23].

There are some limitations to our survey that should be noted. Firstly, the demographic data indicates that most participants came from urban centers, which means that representation from rural and island areas is limited. Additionally, not all Greek hospitals were included in the survey due to challenges in communication. The questionnaire itself also has a lengthy completion time, which may have discouraged some potential participants or caused them to interrupt the process. The lack of open-ended questions could have deprived the participants from adding other barriers that have not been identified in previous studies. Furthermore, participation in the study requires familiarity with or access to technology. It is also important to note that we were unable to determine an adequate sample size, as the total number of physiotherapists working in Greek ICUs is unknown.

To improve our practices, we need to investigate the effectiveness of proposed strategies to overcome the noted barriers, especially hemodynamic instability and incoherence with the ventilator. Besides dedicated physiotherapists, the incorporation of mobility teams or mobility champions could lead to an increase in the percentage of patients mobilized out of bed. But firstly, we need to have a better picture of this, and a point prevalence study on EM is needed. Additionally, although nurses are not usually involved in EM in Greece, we should thoroughly investigate whether we have a change in attitudes and practices over the years.

## 5. Conclusions

Most Greek physiotherapists believe that early mobilization is crucial for the rehabilitation of critically ill patients. A significant percentage know guidelines, yet they do not follow a specific protocol. Various barriers prevent its implementation, which depends on the patients, healthcare providers, and the overall process. Yet, it is recognized that practices and perceived barriers are influenced by experience and work schedule. Establishing clinical protocols is essential to facilitate the implementation of early mobilization and support patient rehabilitation. Future efforts should focus on designing strategies and EM protocols for physiotherapy in Greek ICUs. Also, we need to monitor changes in perceived barriers across other countries as focus on the matter via published studies and clinical seminars could lead to significant changes.

## Figures and Tables

**Figure 1 healthcare-13-01248-f001:**
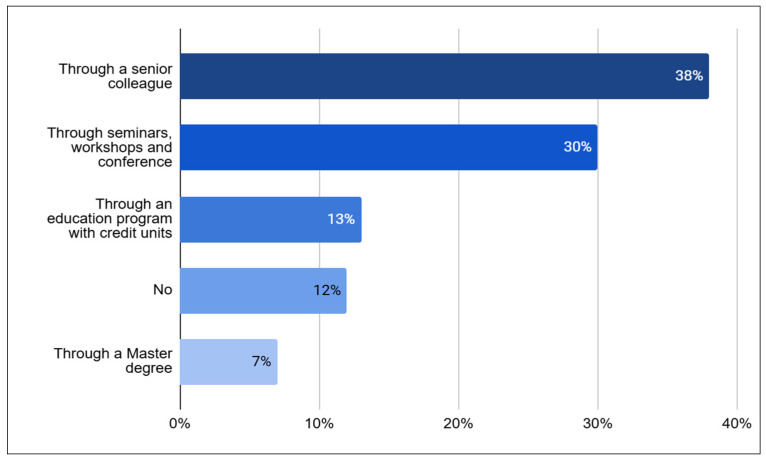
Participants’ training resources.

**Figure 2 healthcare-13-01248-f002:**
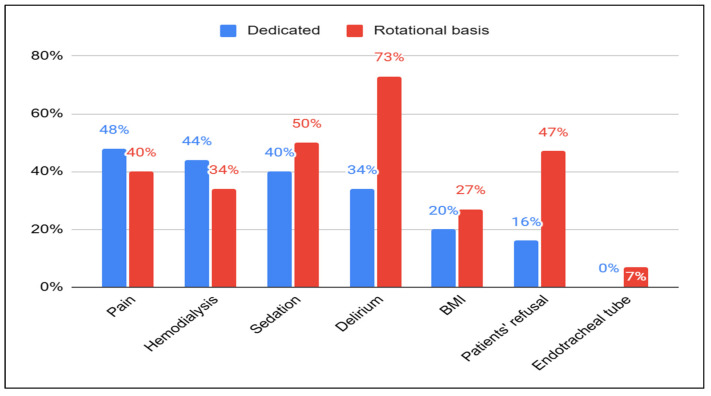
Perceived barriers to EM between dedicated ICU physiotherapists and those working on a rotational basis.

**Figure 3 healthcare-13-01248-f003:**
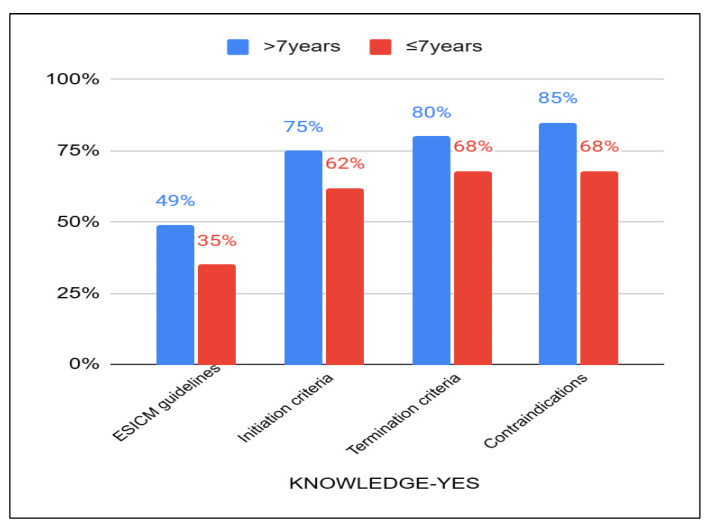
Knowledge of guidelines and EM safety criteria to years of experience between those with >7 years and ≤7 years.

**Figure 4 healthcare-13-01248-f004:**
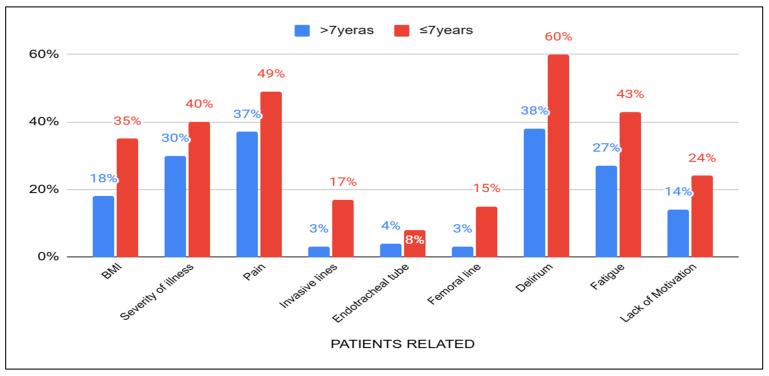
Acknowledgement of patients related barriers to years of experience between those with >7 years and ≤7 years.

**Figure 5 healthcare-13-01248-f005:**
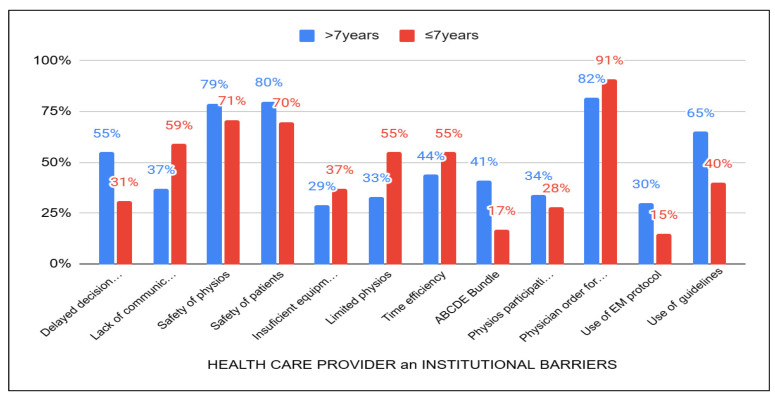
Acknowledgement of healthcare provider and institutional-related barriers to years of experience between those with >7 years and ≤7 years.

## Data Availability

The original contributions presented in this study are included in the article/Appendix A. Further inquiries can be directed to the corresponding author.

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
