# Peer review of "Early Mobilization of Critically Ill Patients: A Survey of Knowledge, Practices and Perceptions of Greek Physiotherapists"

_healthcare, 2025, doi:10.3390/healthcare13111248_

Round 1
Reviewer 1 Report
Comments and Suggestions for Authors
Thank you for the opportunity to contribute to the peer-review process of your manuscript. Although I see the relevance of your work, I believe there are several areas for improvement before it can be published. Please see below.
Introduction.
- Lines 41-42: “Also, the lack of mobilization has resulted in up to 40% muscle mass loss in the first week and 1% bone density loss per week.” I suggest revising this sentence to say something like “the lack of mobilization has been shown to result in up to…”
- Line 71: Avoid starting a sentence with “and.” Try to merge the two sentences or rephrase the second one.”
Methods.
- Line 77: I think it would be very helpful if you attached the actual survey in the supplementary files.
- Line 99: Instead of “managed to create,” could say “created.”
- Line 106: since you posted it on social media, how did you ensure that only those meeting your eligibility criteria responded to the survey?
- Lines 113-114: Describe what was the process for handling their responses and personal information.
Results.
- Line 121: Remove the “a” before 42%.
- Line 122: Choose one piece of information about the age category to report on.
- Line 125: You said “most of them (43%)…” If it is less than 50%, you cannot say “most of.” Suggest saying “the largest percentage of” or something similar.
- Lines 127-128: “75% noted that didn’t have a daily afternoon shift, but 62% had a 127 weekend shift.” This sentence is confusing because “but” suggests contrasting ideas, which does not seem to be the case. Additionally, avoid using contractions such as “didn’t” in academic writing.
- Line 131: Figure 1 would work better as a bar graph ordered from the most to least used training source. Also, is this the most relevant information in this section? It seems to me that there are other things more relevant to your results, but when you create a figure, you draw more attention to it than to the rest that is presented in the text.
- Lines 139-141: The way this is written suggests that supplemental table 2 will be the ESICM guidelines. Suggest re-writing for clarity.
- Lines 143-144: This is confusing because ‘protocol’ and ‘guidelines’ are similar terms, so it appears you are discussing the same concept, but the results obviously add up to more than 100%. I had to refer to the table to clarify, but it won’t be part of the body of your manuscript (it will be included as a supplementary file). So, I suggest re-writing for clarity.
- Line 178: Same as the comment above, 45% is not “most of.” Suggest revising throughout the paper.
- Lines 182-185: This paragraph does not seem to belong here. It appears that you are referring to knowledge (section 4.2) but with different numbers from those you have already reported. If you want to discuss the differences between those dedicated to ICU only and those who work on a rotation scale, that needs to be very clear.
- Lines 186-214: I actually think that the following paragraphs would be way easier to understand if they were reported in a graphic or table. It is too hard to follow the text.
- Line 216: Why did you choose seven years as your threshold? Usually, people would select full numbers, such as 10 years. It gives the impression that you purposefully chose this to support the differences in your findings.
- Line 225: Figure 2 legends should be capitalized.
- Lines 226 onwards: Figures 3 and 4 should have a logical order. Try to report your barriers from most to least prevalent ones (at least in one of the groups). The presentation is not logical or easy to follow.
- A general comment for all your results: You must not write the same thing that is in your tables and figures in the text. It is okay to highlight a few of the more relevant findings in the text, but the way you wrote it is very repetitive. The fact that the tables are supplementary files should not be an excuse. Try different ways to present the same information. Additionally, there are numerous formatting errors in all your tables, including extra spaces, improper punctuation after questions, and inconsistent capitalization. Some of these also happen in the text (e.g., line 345).
Discussion.
- The discussion was somewhat challenging to follow because the results were not presented clearly; however, I did not notice any major flaws. I would suggest adding the lack of open-ended questions as a limitation of your methods, as participants were unable to suggest any barriers beyond the ones you listed.
All comments related to the quality of the employment of the English Language were described in the previous comment to the authors.
Author Response
Reviewer 1.
We would like to thank the reviewer for looking into our work and allowing us to improve our paper through his/her constructive comments. All changes are highlighted in the manuscript.
Comment:
Introduction.
- Lines 41-42: “Also, the lack of mobilization has resulted in up to 40% muscle mass loss in the first week and 1% bone density loss per week.” I suggest revising this sentence to say something like “the lack of mobilization has been shown to result in up to…”
- Line 71: Avoid starting a sentence with “and.” Try to merge the two sentences or rephrase the second one.”
Response. We rephrased all mentioned lines. Specifically lines 41-42 into: “…the lack of mobilization has been shown to result in up to 40% muscle mass loss in the first week and 1% bone density loss per week” as the reviewer suggested. And we merged the two sentences form line 71 into “ To date, there hasn’t been a study examining the barriers to implementing early mobilization which act as inhibitors in promoting the recovery of critically ill patients in the ICU”
Comments
Methods.
- Line 77: I think it would be very helpful if you attached the actual survey in the supplementary files.
Response. The survey was conducted electronically. But, in order to present the survey wel included the questions in the supplemental file along with the responses. If there is a need to add another file with the survey we could do so, but there aren’t other information to be added.
- Line 99: Instead of “managed to create,” could say “created.”
Response. We made the suggested change
- Line 106: since you posted it on social media, how did you ensure that only those meeting your eligibility criteria responded to the survey?
Response. We posted it on specific groups of social media of ICU physiotherapists or physiotherapists in general. But to be certain that the participants work in ICU questions regarding type of work status or years of ICU experience or even the type of ICU could help us exclude physiotherapists that don’t meet our eligibility criteria.
Lines 113-114: Describe what was the process for handling their responses and personal information.
Response. The participation was anonymous. Information that could lead to identification such as the name of their hospital was n’t included. All answers are stored in the platform and access to these have only the ones who created the only survey Dr Patsaki and Ms Papadimitriou. The responses are extracted from the platform in an excel file and this was only shared among those who handled the statistics.
Comments
Results.
- Line 121: Remove the “a” before 42%.
Response. We removed it.
- Line 122: Choose one piece of information about the age category to report on.
Response. we chose to keep the largest percentage “44% (the largest percentage) were 40-49 years old”
- Line 125: You said “most of them (43%)…” If it is less than 50%, you cannot say “most of.” Suggest saying “the largest percentage of” or something similar.
Response. we made the appropriate change. “The largest percentage of participants”
- Lines 127-128: “75% noted that didn’t have a daily afternoon shift, but 62% had a 127 weekend shift.” This sentence is confusing because “but” suggests contrasting ideas, which does not seem to be the case. Additionally, avoid using contractions such as “didn’t” in academic writing.
Response. We changed “but” to “and” in order to avoid confliction and made appropriate changes related to contractions to the whole manuscript.
- Line 131: Figure 1 would work better as a bar graph ordered from the most to least used training source. Also, is this the most relevant information in this section? It seems to me that there are other things more relevant to your results, but when you create a figure, you draw more attention to it than to the rest that is presented in the text.
Response. As the reviewer pointed out when creating a figure we draw attention to this and that is exactly what we wanted. To underline the different training sources that are available in Greece related to the ICU. These support also the difference in the responses related to older similar studies. Continuous training and lifelong learning is most important in our profession and to be able to present an aspect of this is significant. The figure was changed into bars.
- Lines 139-141: The way this is written suggests that supplemental table 2 will be the ESICM guidelines. Suggest re-writing for clarity.
Response. This is true and concerns all supplemental material. Thus we moved All notifications for the supplemental materials to the sub-headings.
- Lines 143-144: This is confusing because ‘protocol’ and ‘guidelines’ are similar terms, so it appears you are discussing the same concept, but the results obviously add up to more than 100%. I had to refer to the table to clarify, but it won’t be part of the body of your manuscript (it will be included as a supplementary file). So, I suggest re-writing for clarity.
Response. A protocol of early mobility is quite different from guidelines on the subject. Guidelines are published by a well established society like the European Society of Intensive Care Medicine. Whilst a protocol is a detailed and structured plan that outlines how to implement a procedure. Thus each hospital could have its own. That is why we included two separate questions.
- Line 178: Same as the comment above, 45% is not “most of.” Suggest revising throughout the paper.
Response. Appropriate changes were made.
- Lines 182-185: This paragraph does not seem to belong here. It appears that you are referring to knowledge (section 4.2) but with different numbers from those you have already reported. If you want to discuss the differences between those dedicated to ICU only and those who work on a rotation scale, that needs to be very clear.
Response. The paragraph belongs to a subsection of results 4.7 work schedule variances. We state differences between those who work on a rotation scale and those dedicated to work only in the ICU. And this is clarified with the first sentence of the sub-section “We observed variances in Greek physiotherapists' knowledge, practices and perceived barriers related to their ICU work schedules”.
- Lines 186-214: I actually think that the following paragraphs would be way easier to understand if they were reported in a graphic or table. It is too hard to follow the text.
Response. In order to better present the results as we did with the years of experience and as suggested by the reviewer, we added another figure (Figure 2) regarding perceived barriers among physiotherapists with different work schedules. Perceived barriers present most differences.
- Line 216: Why did you choose seven years as your threshold? Usually, people would select full numbers, such as 10 years. It gives the impression that you purposefully chose this to support the differences in your findings.
Response. In our survey years of experience was divided in 3 categories: 3 years, 3-7 years and >7 years. We included 3 years to include new colleagues that were employed in the ICU during the COVID pandemic. And while looking into relevant studies, we noticed that they used less years without finding any differences. For example Jolley et al 2014 (ref 21) used 5years as a cut off point and found no differences, Fontela et al 2018 (Ref9) examined both nurses and physiotherapists again using the 5 years cut off point and again found no difference. What led us also to use the 7 years is that there are studies that measured years of experience and identified that 7 years is the mean value (Barber et al 2015;Lago et al 2022). And, there is a variance regarding this for example Alqahtani et al.,2020 used 1 – 3,4 -6 ,7- 8,9 – 10 , >10years and Paulo et al., 2021 used :<5 , 6 – 10 , 11- 15 , 16- 20 , >20 years.
- Line 225: Figure 2 legends should be capitalized.
Response. Changes were made
- Lines 226 onwards: Figures 3 and 4 should have a logical order. Try to report your barriers from most to least prevalent ones (at least in one of the groups). The presentation is not logical or easy to follow.
Response. We followed the reviewers recommendation and we rearranged findings in the text from most to least prevalent.
- A general comment for all your results: You must not write the same thing that is in your tables and figures in the text. It is okay to highlight a few of the more relevant findings in the text, but the way you wrote it is very repetitive. The fact that the tables are supplementary files should not be an excuse. Try different ways to present the same information. Additionally, there are numerous formatting errors in all your tables, including extra spaces, improper punctuation after questions, and inconsistent capitalization. Some of these also happen in the text (e.g., line 345).
Response. We went through all supplemental tables and made appropriate corrections. The supplemental tables have all questions that were included in the survey ( so the reader we have the whole picture of what was asked and all the answers). In text and in figures we chose the most significant to present. We may have repeated findings in text and in figures, but we believe that a figure could better highlight the extent of these differences. Also taking into consideration that we believe that such as extensive work on the matter and especially looking into differences among different work schedules and years of experience should be fully presented as we haven’t noticed another similar one.
Discussion.
- The discussion was somewhat challenging to follow because the results were not presented clearly; however, I did not notice any major flaws. I would suggest adding the lack of open-ended questions as a limitation of your methods, as participants were unable to suggest any barriers beyond the ones you listed.
Response. Although the survey and all included barriers were based on previous relative studies, we included the barrier suggested by the reviewer (lines 329-331).

Reviewer 2 Report
Comments and Suggestions for Authors
Article: Early mobilization of critically ill patients: A survey of Knowledge, Practices and Perceptions of Greek physiotherapists.
Thank you for giving me the opportunity to evaluate such a great research paper.
It is a great honor to have the opportunity to review such a wonderful paper: Early mobilization of critically ill patients: A survey of Knowledge, Practices and Perceptions of Greek physiotherapists. Let me comment on some of the contents of the paper.
- Page 1,
"Early mobilization of critically ill patients: A survey of knowledge, practices, and perceptions of Greek physiotherapists"
This study is very meaningful as it clearly shows the current status of Greek physiotherapists' work and the direction for improvement.
- Page 2 Introduction
It is good to have a clear purpose for the research. And it is important that the clear purpose is described in a way that shows sufficient direction as a research. Review the content of the research purpose article to see if it is clear.
- Page 2 Materials and Methods
There are many contents in SPSS statistical methods. Is there a reason why SPSS was used when professional analysis methods are not used? Looking at the analyzed data, general contents were analyzed, so is there a reason why Excel was not used? Since SPSS is used, it must have been done because professional statistical analysis methods were used, but is there anything that was not described additionally? Please review the differences between previous studies and this study regarding the statistical methods required for the questionnaire analysis method.
- Page 2 Materials and Methods
The questionnaire in the appendix contains a lot of important content. However, there is no explanation for this content. It is necessary to mention the content of each item in the method by category. Please summarize the content needed for analysis for each item by describing why it is needed in the questionnaire.
- Page 3~11 Result
The content of the results is very useful for readers. To do so, it is important that the content of the results is well organized. I think that the data that the researcher has statistically analyzed is important as visual data. Therefore, it would be better to show the analyzed graph and description together. Please insert more graphs necessary for the description and organize the results into an easy-to-understand content.
It is a great honor to review research.
Please write a little more so that the good research results can be easily read.
Thank you.
sincerely
Author Response
Reviewer 2.
We are most thankful for the reviewer’s overall comments on our manuscript and suggestions made to help us improve our work.
- Comment.Page 1,"Early mobilization of critically ill patients: A survey of knowledge, practices, and perceptions of Greek physiotherapists"
This study is very meaningful as it clearly shows the current status of Greek physiotherapists' work and the direction for improvement.
Response. Thank you for the comment. The recognition of our work and effort to present Greek physiotherapists’ involvement in the field of ICU practice.
- Comment. Page 2 Introduction
It is good to have a clear purpose for the research. And it is important that the clear purpose is described in a way that shows sufficient direction as a research. Review the content of the research purpose article to see if it is clear.
Response. In order to clarify our aim, we rephrased lines 72-73 into “The primary aim of the present study was to identify physiotherapists' knowledge and practices regarding EM. It was also to identify their perceptions of barriers to EM in Greek ICUs.”
- Comment. Page 2 Materials and Methods
There are many contents in SPSS statistical methods. Is there a reason why SPSS was used when professional analysis methods are not used? Looking at the analyzed data, general contents were analyzed, so is there a reason why Excel was not used? Since SPSS is used, it must have been done because professional statistical analysis methods were used, but is there anything that was not described additionally? Please review the differences between previous studies and this study regarding the statistical methods required for the questionnaire analysis method.
Response. Although at the beginning Excel was used as all answers were exported in such a format, we decided to use the SPSS in order to be able to separate the participants in different groups to perform further analysis. The statistics used, even simple ones, are widely used in previous studies ref 13,14, 16, 17, 25. As our purpose was to describe and note possible variance we believe that with the descriptive statistics we were able to fulfill our aim.
- Comment. Page 2 Materials and Methods
The questionnaire in the appendix contains a lot of important content. However, there is no explanation for this content. It is necessary to mention the content of each item in the method by category. Please summarize the content needed for analysis for each item by describing why it is needed in the questionnaire.
Response. In the appendix we included tables with the answers from the domains of the survey presented at the section of materials and methods. (lines 85,86) More specifically we have the following domains: participants' knowledge of early mobilization (6 items)-answers presented in suppl. table 1., practices (7 items)- answers presented at suppl table 2., perception on EM (11 items)- answers presented at suppl table 3 and perceived barriers (29 items) to its implementation- answers presented at suppl table 4. As the information was quite a lot, we decided to summarize findings in the text and present all the questions along with the answers at the appendix. Like this we could share the whole survey and the answers given.
- Comment.Page 3~11 Result
The content of the results is very useful for readers. To do so, it is important that the content of the results is well organized. I think that the data that the researcher has statistically analyzed is important as visual data. Therefore, it would be better to show the analyzed graph and description together. Please insert more graphs necessary for the description and organize the results into an easy-to-understand content.
Response. It is true that there are a lot of data to be presented. We use subtitles to organize our results. We decided to add certain figures for the results that were most important and we included an additional figure for differences in perceived barriers related to EM in relation to work schedule (Figure 2).We tried to rearrange text and graphs in order to be closer together. We avoided further figures in order to avoid repetition as pointed out by another reviewer.

Reviewer 3 Report
Comments and Suggestions for Authors
Dear authors,
Thank you for your submission. Below, I will specify some relevant aspects to consider point by point:
- Lines 5-6: Insert "*" next to the corresponding author whose e-mail is in line 9.
- Abstract: Verify that all abbreviations are defined upon their first use and are used consistently throughout the text (e.g., EM, ICU, BMI…).
- Line 35: Write "Intensive Care Units" instead of "Intensive Care Unit", "Perception" instead of "Perceptions", and "Physical Therapists" instead of "Physiotherapists" to ensure these keywords align with MeSH terms. "Early mobilization" and "Barriers" are not MeSH terms but are acceptable as they are topic-specific terms.
- Line 38: Write "Intensive Care Unit" instead of "Intensive Unit Care".
- Line 83: It is important to specify which sociodemographic variables, such as sex, age, or other relevant details, were collected in the questionnaire and how they were gathered, as this information is not included in the supplemental tables.
- Statistics: Write "2.3 Statistics" instead of "3. Statistics". Specify the version of SPSS used. Define the types of charts presented in the figures, and highlight the distribution of the two groups (more than seven years and less than seven years of work experience in the ICU) in Figures 2, 3, and 4.
- Results: Several details from section 4.1 "Participants" are repeated later in the "Results" section, specifically after the phrases "as mentioned earlier..." (lines 176 and 216). Please avoid including redundant information.
- Line 136: Figure 1 should be mentioned before its appearance in the manuscript.
- Line 150: Write "3.4. Perceptions" instead of "4.5. Perceptions".
- Lines 182, 186, and 195: Replace the repetitive phrase "Regarding..." with alternatives like "Concerning..." or "In terms of...".
- Lines 197 and 310: The abbreviations for "Body Mass Index" (BMI) and "The European Respiratory Society" (ERS) are unnecessary as these terms appear only once in the manuscript.
- Lines 293-295: Use an appropriate source (article, website, or other resource) to justify the idea. You may refer to the previously cited references.
- Lines 344-345: Specify the principal barriers identified in the manuscript that hinder the implementation of early mobilization.
- Line 384: Please delete this line to maintain correct reference numbering.
- References: Ensure compliance with reference formatting guidelines, checking for any typographical errors. If possible, include the volume number for each article, not just the issue number.
Based on this information, I recommend that the manuscript be revised and rewritten.
Best wishes.
Comments on the Quality of English LanguageThe ideas in the text are clear, but sometimes, the scientific tone is weakened through informal expressions and words that could cause confusion.
Author Response
Reviewer 3.
We would like to thank the reviewer for taking the necessary time and effort to review our manuscript. We sincerely appreciate all your valuable comments.
- Comment. Lines 5-6: Insert "*" next to the corresponding author whose e-mail is in line 9.
Response. This is added.
- Comment. Abstract: Verify that all abbreviations are defined upon their first use and are used consistently throughout the text (e.g., EM, ICU, BMI…).
Response. We went through the manuscript and searched for abbreviations that weren’t defined like ICUand BMI in the abstract as the reviewer kindly pointed out. All are highlighted.
- Comment. Line 35: Write "Intensive Care Units" instead of "Intensive Care Unit", "Perception" instead of "Perceptions", and "Physical Therapists" instead of "Physiotherapists" to ensure these keywords align with MeSH terms. "Early mobilization" and "Barriers" are not MeSH terms but are acceptable as they are topic-specific terms.
Response. All suggested changes are made.
- Comment. Line 38: Write "Intensive Care Unit" instead of "Intensive Unit Care".
Response. The change is made.
- Comment. Line 83: It is important to specify which sociodemographic variables, such as sex, age, or other relevant details, were collected in the questionnaire and how they were gathered, as this information is not included in the supplemental tables.
Response. We included sex, age, residential area (urban, rular etc) and educational level. These were included as introductory questions before the main domains of the survey. We included these information at Methods and subsection 2.1. (lines 86-87) Data on these sociodemographic variables are included in the results, we avoided adding an extra supplemental table as main results are given in text.
- Statistics: Write "2.3 Statistics" instead of "3. Statistics". Specify the version of SPSS used. Define the types of charts presented in the figures, and highlight the distribution of the two groups (more than seven years and less than seven years of work experience in the ICU) in Figures 2, 3, and 4.
Response. Suggested changes are made. We specified the version of SPSS used (21) and we highlighted the distribution of the two groups regarding years of experience.
- Results: Several details from section 4.1 "Participants" are repeated later in the "Results" section, specifically after the phrases "as mentioned earlier..." (lines 176 and 216). Please avoid including redundant information.
Response. The numbers of sections and subsections have changed in relation to the change made to Statistics. The subsection of “Participants” presents information and data related to all the participants of the survey. Then at subsections 3.6 and 3.7 we present information and data on two separate groups in relation to the years of experience and the type of work schedule. We don’t repeat all data, yet we need to underline the percentage of the participants included in each group examined.
- Line 136: Figure 1 should be mentioned before its appearance in the manuscript.
Response. We changed the position of figure 1.
- Line 150: Write "3.4. Perceptions" instead of "4.5. Perceptions".
Response. Sorry for the typographic errors. We corrected them and changed the order according to a previous comment.
- Lines 182, 186, and 195: Replace the repetitive phrase "Regarding..." with alternatives like "Concerning..." or "In terms of...".
Response. We appreciate all comments that will improve English writing.
- Lines 197 and 310: The abbreviations for "Body Mass Index" (BMI) and "The European Respiratory Society" (ERS) are unnecessary as these terms appear only once in the manuscript.
Response. Abbreviations have been erased.
- Lines 293-295: Use an appropriate source (article, website, or other resource) to justify the idea. You may refer to the previously cited references.
Response. We added relevant ref for a prior Greek study (ref 16).
- Lines 344-345: Specify the principal barriers identified in the manuscript that hinder the implementation of early mobilization.
Response. We added the principal barriers identified.
- Line 384: Please delete this line to maintain correct reference numbering.
Response. We removed the line-probably was introduced when placing text into the journals’ format as we don’t have ref 33. Yet we checked the references again.
- References: Ensure compliance with reference formatting guidelines, checking for any typographical errors. If possible, include the volume number for each article, not just the issue number.
Response. In accordance to the journals format, volume numbers are not included only the number of issue. We looked again through the reference to avoid any typographical errors.

Round 2
Reviewer 3 Report
Comments and Suggestions for Authors
Thank you for resubmitting the article. The manuscript has shown substantial improvement in quality. However, there are two important points that still need to be addressed:
- In the Statistics section, please indicate that bar graphs are used to present the data. Including this detail will enhance clarity and transparency in your description of the data presentation methods.
- Figures 3, 4, and 5: Please revise these figures to ensure consistency. All labels should begin with capital letters, and any orthographic errors should be corrected.
Once these revisions are made, I believe the manuscript will be suitable for publication.
Best regards.
Author Response
We are most thankful for the reviewer’s overall comments on our manuscript and suggestions made to help us improve our work.
Comments
- In the Statistics section, please indicate that bar graphs are used to present the data. Including this detail will enhance clarity and transparency in your description of the data presentation methods.
- Figures 3, 4, and 5: Please revise these figures to ensure consistency. All labels should begin with capital letters, and any orthographic errors should be corrected.
Responses.
- In the statistics subsection we included the following: “ Descriptive statistics were used to summarize the responses for all items, along with bar graphs to present our findings.”
- We went through the figures noted by the reviewer and made all appropriate changes.
